# Limited Phenotypic Variation in Vulnerability to Cavitation and Stomatal Sensitivity to Vapor Pressure Deficit among Clones of *Aristotelia chilensis* from Different Climatic Origins

**DOI:** 10.3390/plants10091777

**Published:** 2021-08-26

**Authors:** Marco A. Yáñez, Javier I. Urzua, Sergio E. Espinoza, Victor L. Peña

**Affiliations:** 1Instituto de Investigación Interdisciplinaria, Vicerrectoría Académica, Universidad de Talca, 2 Norte 685, P.O. Box 747, Talca 3460000, Chile; 2Universidad de Talca, 2 Norte 685, P.O. Box 747, Talca 3460000, Chile; javier.urzua@utalca.cl; 3Facultad de Ciencias Agrarias y Forestales, Universidad Católica del Maule, Talca 3460000, Chile; espinoza@ucm.cl; 4Forestry Consultant, 8 Norte and 22 Oriente 3337, Talca 3460000, Chile; vpenaf9@gmail.com

**Keywords:** clonal variation, drought tolerance, plant adaptation, xylem embolism, acclimation

## Abstract

*Aristotelia chilensis* (Molina) Stuntz is a promising species in the food industry as it provides ‘super fruits’ with remarkable antioxidant activity. However, under the predicted climate change scenario, the ongoing domestication of the species must consider selecting the most productive genotypes and be based on traits conferring drought tolerance. We assessed the vulnerability to cavitation and stomatal sensitivity to vapor pressure deficit (VPD) in *A. chilensis* clones originated from provenances with contrasting climates. A nursery experiment was carried out for one growing season on 2-year-old potted plants. Measurements of stomatal conductance (gs) responses to VPD were taken in spring, summer, and autumn, whereas vulnerability to cavitation was evaluated at the end of spring. Overall, the vulnerability to cavitation of the species was moderate (mean P_50_ of −2.2 MPa). Parameters of the vulnerability curves (*K_max_*, P_50_, P_88_, and S_50_) showed no differences among clones or when northern and southern clones were compared. Moreover, there were no differences in stomatal sensitivity to VPD at the provenance or the clonal level. However, compared with other studies, the stomatal sensitivity was considered moderately low, especially in the range of 1 to 3 kPa of VPD. The comparable performance of genotypes from contrasting provenance origins suggests low genetic variation for these traits. Further research must consider testing on diverse environmental conditions to assess the phenotypic plasticity of these types of traits.

## 1. Introduction

Many forest ecosystems are threatened by higher temperatures and more intense droughts due to climate change [1]. It is not well understood how different species will adapt to this new scenario, which may imply potential shifts in the species distribution or mortality induction. Thus, understanding local adaptation to climate in properties associated with drought resistance may be a way of selecting and managing specific genotypes for commercial purposes, and also of assessing the potential impact of climate change on a given species [2,3].

Hydraulic traits are essential to understand plant responses to drought [4]. To maintain the plant water status, plants exert stomatal control to regulate water loss by transpiration and counteracting soil water deficit and evaporative demand (i.e., vapor pressure deficit, VPD) [5]. The tension to drive water movement within a plant generates a metastable state under certain thresholds and promotes embolisms, decreasing the hydraulic conductivity [6]. Vulnerability to cavitation is one of the most important characteristics defining drought-induced tree mortality [7,8,9]. Variation in vulnerability to cavitation has been studied at the species level [10,11]. However, intraspecific variation (i.e., genetic differentiation among provenances) in cavitation resistance has received little attention and has shown contrasting results. Differences among provenances in vulnerability curve parameters have been found in *Pseudotsuga menziesii* [6], *Betula pendula*, *Populus tremula*, *Picea abies, Pinus sylvestris* [12], and *Pinus pinaster* [13], but no differences have been reported for species such as *Pinus hartwegii* [14] and *Picea glauca* [3]. Similarly, significant intrapopulation variation (i.e., clonal level) has been reported in vulnerability curve parameters in *Populus nigra* L. [15] but not in *Hevea brasiliensis* [16], *Populus tremuloides,* and poplars [17]. Thus, the extent of genetic differentiation between and within populations in vulnerability seems to vary among plant species.

Without consideration of the soil water availability, plant water status depends on the evaporative demand, and stomata respond to the temporal variation in VPD [18], affecting carbon assimilation and plant growth. Stomatal conductance (*g_s_*) typically decreases with increased VPD, and the magnitude of the decline (i.e., the slope of the relationship) is termed ‘stomatal sensitivity’ [19]. Stomatal sensitivity to VPD has been shown to be directly related to *g_s_* at low VPD (i.e., *g_ref_*) [19,20]. Some studies have shown differences in stomatal sensitivity to VPD between and within plant species [18,21,22,23], whereas others have not [24]. Overall, higher stomatal sensitivity to VPD has been found in species from humid climate origins compared with those from arid climate origins and associated with warmer temperatures [23,25]. However, little is known about how climate origin may exert genetic differentiation in stomatal sensitivity to VPD among populations of a given species.

*Aristotelia chilensis* (Molina) Stuntz, commonly named ‘maqui’, is a native species in Chile that belongs to the Elaeocarpaceae family. The species’ range of distribution spans over 15° latitude and harbors a wide range of environmental conditions from Mediterranean semiarid to temperate subhumid and humid climates [26,27]. Its red/purple berries are known to have high antioxidant activity and anthocyanin content, even compared with some other well-known berries, such as blueberries (*Vaccinium corymbosum)*, pomegranates (*Punica granatum*), blackberries (*Rubus fruticosus*), raspberries (*Rubus idaeus*), and cranberries (*Vaccinium oxycoccos*) [28,29]. Regarding the idea that the species may play an important role in future food security [30], there is some ongoing research about its domestication, and some clonal varieties have been deployed for commercial purposes [31]. On the other hand, regarding the broad range of environmental conditions in which the species develops, a high phenotypic plasticity and genetic variation are expected for important functional traits related to drought tolerance, but this is still poorly understood. In Mediterranean zones, under a climate change scenario, the success of a future genotype selection must consider traits related to drought tolerance. In this study, we assessed the vulnerability to cavitation and stomatal sensitivity to VPD of *A. chilensis* clones originated from provenances with contrasting climates, which, to the best of our knowledge, is the first data set produced for this species. We expected that the differences in environmental conditions of origin clones were translated into phenotypic variability in cavitation resistance and stomatal sensitivity to VPD. Understanding the variation of this important key fitness trait might contribute to the ongoing domestication process and know the species’ genetic variation to conduct water through the xylem even during extreme drought events.

## 2. Results

There were no significant differences among clones in the vulnerability curve parameters (P_50_: F_3.10_ = 0.69, *p* = 0.5780, P_88_: F_3.10_ = 0.52, *p* = 0.6224, S_50_: F_3.10_ = 1.11, *p* = 0.3901, *K_max_*: F_3.10_ = 0.43, *p* = 0.7340). Similarly, there were no differences among the genotypes from the southern versus the northern provenances (P_50_: F_1.10_ = 0.94, *p* = 0.3540, P_88_: F_1.10_ = 0.85, *p* = 0.4450, S_50_: F_1.10_ = 2.54, *p* = 0.1420, *K_max_*: F_1.10_ = 0.72, *p* = 0.4151). Thus, under no genetic differentiation, the curve with the pooled data describing the species (Figure 1) had parameters of P_50_ = −2.2 MPa, P_88_ = −3.95, S_50_ = 46.6%, and *K_max_* = 8.8 mmol s^−1^ MPa^−1^.

The ranges for VPD and *g_s_* measurements were 0.86 to 4.7 kPa and 0.02 to 0.43 mol m^−2^ s^−1^, respectively. There was great variability of *g_s_* across the VPD range, which was relatively uniform in the range of 1 to 3 kPa, and then it decreased (Figure 2). As expected, increasing VPD significantly decreased *g_s_* (Figure 2A provenance model: F_1.162_ = 35.23, *p* < 0.0001; Figure 2B clonal model: F_1.162_ = 38.32, *p* < 0.0001). The response of *g_s_* to VPD did not differ among provenances, as these exhibited the same slope (VPD × Provenance: F_2.162_ = 0.325, *p* = 0.7230) and Y-intercept (VPD × Clone: F_2.162_ = 0.0110, *p* < 0.8960) (Figure 2A). However, at the clonal level we found a similar slope (VPD × Provenance: F_9.148_ = 0.4790, *p* = 0.8872) but different Y-intercept (VPD × Clone: F_9.148_ = 2.445, *p* = 0.0127) (Figure 2B). Then, the average slope was −0.082, and the reference stomatal conductance (*g_ref_*) at 1 kPa of VPD was 0.277 mol m^−2^ s^−1^ (m-to-*g_ref_* ratio was 0.3). The average *g_s_* for the study period was not related to provenance origin and was significantly higher for clone A_320 from provenance SanFer, whereas the lowest values were observed for clones A_319 (SanFer provenance) and F_523 (Enlagos provenance) (Figure 2B inset graph).

## 3. Discussion

Studying the genotypic variation of key traits related to drought tolerance may be a way to understand the mechanisms to cope with extreme drought events and predict forest species’ adaptation potential to climate change in Mediterranean-type climate areas. In general, comparisons among studies on vulnerability to cavitation are difficult because of the differences in sampling strategies and measuring techniques for its evaluation [32,33]. Regarding the idea that *A. chilensis* is a pioneer species that grows in a wide variety of soils and climate conditions [30], we expected to find phenotypic variability in vulnerability to cavitation as an adaptive trait for drought tolerance that could explain the species’ success in those diverse environments. Overall, *A. chilensis* has been shown to synthesize anthocyanins with high antioxidant activity as a mechanism to tolerate drought stress [2]. However, other traits, such as stomatal sensitivity and hydraulic conductivity, might better explain the local adaptations of plant species to drought [34]. Our study showed that vulnerability to cavitation and stomatal sensitivity to VPD did not differ among *A. chilensis* genotypes originated from contrasting climatic conditions (provenances from the center and southern parts of the species’ distribution), and that compared with other commercial forest species, *A. chilensis* has a moderate vulnerability to cavitation and moderately low stomatal sensitivity to VPD.

Clones from the Enlagos provenance come from wetter environments, with lower mean temperature and radiation, compared with clones from the SanFer and Romer provenances (Table 1). We expected that the differences in environmental conditions in which *A*. *chilensis* grows might have imposed adaptive pressures over the populations under study in traits conferring drought tolerance. Although this study included only a few provenances and genotypes, these corresponded to clonal material grown under well-controlled environmental conditions, which allowed us to assess the genotypic variability more precisely in the traits under study [35]. The variation in vulnerability to cavitation between populations seems to be species dependent. Some studies have shown variation among populations [6,12,13], whereas others have not [3,14,35]. Similarly, controversial results have been found at the clonal level [15,16,17]. The low genetic differentiation in vulnerability to cavitation among natural populations might indicate ‘uniform selection’ for this attribute rather than genetic drift, as has been suggested for other forest species [35,36,37]. However, we still have to evaluate the full range of provenances to test this hypothesis in *A*. *chilensis*, but this assertion might be supported by the findings of Salgado et al. [38], who found no neutral genetic differentiation among *A. chilensis* populations, including the populations in our study. Additionally, González-Muñoz et al. [12] mentioned that the low variation in the cavitation parameters might also suggest low potential hydraulic adaptation of species populations to drier conditions. However, our study only includes approximately half of the whole latitudinal range of *A*. *chilensis*. In the northern range of the species’ distribution, with precipitation levels lower than 200 mm year^−1^, *A*. *chilensis* grows as a small shrub, whereas in the southern part of the distribution range, with annual precipitations higher than 1500 mm, the species grows as a tall tree. It is generally accepted that plants with dry climate origins have denser wood and are more cavitation resistant than species from wet origins [4,39,40]. However, as our preliminary results do not confirm an adaptive association between resistance to cavitation and precipitation at the provenance origin (i.e., a proxy of water availability), we are aware that further research is needed with new and accurate techniques to represent the full range of growth conditions found in the northern and southern limits of the species distribution and explore whether this attribute can be used as a selection criterion. Moreover, some studies have reported that response to vulnerability to cavitation of specific genotypes or populations changes with the growing environmental condition [35,41,42], contributing to the phenotypic plasticity of stem hydraulic traits. Under a climate change scenario, the limited genetic variation of vulnerability to cavitation of *A. chilensis* found in our study might be counteracted by a high phenotypic plasticity, which needs to be addressed in future studies.

The mean P_50_ (−2.2 MPa) found in *A. chilensis* was considered moderately high compared with other forest species, such as pines [14], *Fagus sylvatica* [35], *Juniperus communis* [43], *Eucalyptus globulus* [44], and some fruit species, such as *Prunus* spp. [7], apples [45], and walnuts (*Juglans* spp.) [37]. However, the values were slightly lower than those for grapevine varieties [46], *Juglans* spp. [37], and poplars species, the latter considered among the most vulnerable species to drought-induced cavitation in the northern hemisphere [15,47]. Values of P_50_ were similar to those reported for *Taxodium distichum*, which has low resistance to cavitation compared with other conifer species [10,48]. Similarly, the mean P_88_ (−3.95) was high compared with some conifer species, but in the range of angiosperm species, such as *Fagus sylvatica* L., *Populus x canescens* (Aiton) Sm., *Populus tremula* L., *Sorbus torminalis,* and species of the genus *Quercus* [49,50].

The lack of variability between provenances also highlights the genetic limitation of the species to evolve in the hydraulic parameters compared with other phenotypic traits [12]. In this respect, a higher stomatal control to drought events might be a way to reduce the water tension in the xylem, preventing cavitation [51]. In *Eucalyptus* species, Bourne et al. [25] found that whole-canopy stomatal sensitivity to VPD was lower in species from arid climate origins than those from humid climates. Unlike in *A. chilensis*, we found no differences in stomatal sensitivity to VPD (i.e., same slope) among genotypes from differing provenance origins (Figure 2). In our experiment, the substrate moisture content was not limiting because leaf water potentials (ψ_leaf_) were in the range of −0.4 to −0.8 MPa. Thus, the mean sensitivity of *g_s_* to VPD (i.e., m = −0.082) might be interpreted as maximum values expressed by the species under the environmental conditions assessed and not conditioned by leaf water potentials. Differences in stomatal sensitivity to VPD have been found mainly between species [18,19,25], but some evidence has shown genetic differentiation within a species [52], which was not the case in this study. Compared with other studies, our data exhibited great variation of *g_s_* across the VPD range. A uniform variability of *g_s_* could be observed throughout VPD values of 1 to 3 kPa, and then there was a decrease of up to 5 kPa. This pattern suggests a relatively low stomatal sensitivity within the range of 1 to 3 kPa, which is the range observed in the study period (Figure 3), and explains the high productivity of the species when gas exchange is not limited by soil moisture. Otherwise, increases in VPD by over 3 kPa under climate change might strongly reduce *g_s_*, especially under soil water limitations, which need further research. The stomatal conductance found in *A. chilensis* was in the range reported for other berry species, such as highbush blueberry (*Vaccinium corymbosum* L.) [53] and grapevine [54], crops that need irrigation supplies in Mediterranean climates. Moreover, the low m-to-*g_ref_* ratio found in this study (i.e., 0.3) relative to the theoretical ratio (i.e., 0.6) suggested by Oren et al. [19] highlights the anisohydric behavior of *A. chilensis*. It is known that anisohydric species are at a larger risk of xylem cavitation and hence hydraulic failure [55], but this hypothesis needs further research. Regarding the results found in this study, future research must consider a higher number of genotypes and replicates and testing in a more diverse range of environmental conditions (e.g., drought conditions or multiple sites) to assess the phenotypic plasticity of these types of traits, especially at the fruit production stage.

## 4. Materials and Methods

### 4.1. Plant Material and Study Design

In winter 2019, 7 cm long unrooted cuttings of *A. chilensis* plants were collected from a clonal bank belonging to the Centro de Plantas Nativas de Chile (CENATIV), located at the Universidad de Talca, Maule Region, Chile (35°24′ S, 71°38′ W, 112 m. asl). The study considered genotypes of two provenances from the center (SanFer and Romer) and south (Enlagos) of the latitudinal distribution range of the species, which exhibited clear latitude-related climate differences (Table 1) and corresponded to five clones from SanFer, two clones from Romer, and three clones from Enlagos. The plant material was rooted in cold beds filled with perlite and cultivated for 4 months. After that period, the plants were transplanted to 1 L plastic bags filled with local topsoil and grown for 6 more months. During this period, the plants remained in a nursery facility covered with an 80% black polyethylene mesh (Raschel^MR^, Santiago, Chile). Table 1 shows the climatic characteristic of the nursery location.

In winter 2020, part of the plant material was transplanted to new 11 L pots for experiment 1 (cavitation assessment), while the remaining plants were transplanted to 40 L pots for experiment 2 (stomatal sensitivity assessment). We used a completely randomized design (CRD) with six repetitions in a single plant plot in both experiments. The six plants per clone were used for measurements. At this stage, the substrate consisted of a mixture of DSM2 peat (Kekkilä Professional Inc., Vantaa, Finland) and coconut fiber (Golden Grow by Projar, Valencia, Spain) at a volume proportion of 80% and 20%, respectively. In both experiments, the plants were fertilized at the beginning of spring with 12 g of Basacote 9M (Compo Expert GmbH, Münster, Germany) and irrigated daily to pot-substrate capacity using an automatic drip irrigation system. Substrate water content and leaf water potential (ψ_leaf_) were monitored using a ThetaProbe soil moisture sensor (Delta-T Ltd., Cambridge, UK) and a Scholander pressure bomb (PMS Instrument Company, Albany, OR, USA), respectively. During the study period, the plants were well watered to avoid water stress, and the substrate water content varied from 45% to 52%, whereas ψ_leaf_ varied from −0.4 to −0.8 MPa.

### 4.2. Vulnerability Curves

Vulnerability curve analysis was carried out with 2-year-old plant material from experiment 1 and included one clone from the provenance SanFer (A_312), one clone from the provenance Romer (B_219), and two clones from the provenance Enlagos (F_507 and F_523) (i.e., a total of 24 plants). At the end of spring 2020, some stem samples were collected to measure vessel length in some samples using the air method described by Ewers and Fisher [56]. The vessel length ranged from 15.1 to 21.3 cm, with a mean of 18.5 cm. Therefore, we defined a standard length of 20 cm for stem samples for cavitation assessments. Then, stem samples were obtained from all the plants for the genotypes assessed, recut under cold water to a standard length, and kept rehydrated for a whole night. Leaves were removed from the samples to avoid transpiration. Both ends were debarked and carefully trimmed with a razor blade to prevent squeezing of the conduits for the measurements. Xylem vulnerability to cavitation was measured using a Scholander bomb connected to a double-ended pressure chamber (PMS Instrument Company, Albany, OR, USA). The stem samples were mounted on the chamber, with the basal end connected to a tubing apparatus, which had a water reservoir at 60 cm height, and was filled with filtered and degasified water. An opened-ended tube allowed the venting of air bubbles during pressurization, as described by Sperry and Saliendra [57]. The maximum hydraulic conductivity was measured at 6 kPa, and then increased with the pressure bomb at a 0.5 MPa step until reaching 5.5 MPa. After each step pressure, we used 5 min of free flow before measuring the conductance. Then the water flow was collected and immediately weighed in a digital balance with 10 mg precision (Boeco Gmbh, Hamburg, Germany). The percentage loss of conductivity (PLC) was calculated as PLC = 100 × (1 − *K*/*K_max_*), where *K* was the conductance at each pressure step, and *K**_max_* was the maximal conductance measured. The vulnerability curve was fitted for each plant using the reparameterized Weibull function [58,59]. From the model, we derived the parameters P_50_ and P_88_ (estimated the pressure to get 50% and 88% loss of conductivity, respectively) and S_50_ (the slope of the curve in the inflection point).

### 4.3. Stomatal Sensitivity to VPD

To represent the variations in VPD within the second growing season of 2020, the leaf stomatal conductance was measured on three dates at the end of spring, two dates in summer, and one date at the beginning of autumn in experiment 2 (Figure 3). Moreover, measurements were taken preferably in the morning, but we included measurements in the afternoon on some dates to get high VPD values (hourly range between 9 a.m. and 4 p.m.). On the measurement dates, the plants were well watered at a pot-holding capacity. Stomatal conductance (*g_s_*, mmol m^−2^s^−1^) was measured using a portable gas exchange system, LI-6800 (Li-Cor Inc., Lincoln, NE, USA). The initial chamber conditions of temperature and relative humidity were set up at ambient conditions during measurements. In some cases, to get higher VPD values (>3.5 kPa), we controlled these parameters directly in the chamber (VPD_leaf_ function of LI-6800). During the study period, VPD varied from 0.86 to 4.5 kPa. CO_2_ concentration was set at 400 ppm, and PAR at 1800 µmol m^−2^s^−1^. Measurements were taken on fully expanded leaves from the upper third of the plant. The gs response to VPD was fitted as *g_s_* = *g_ref_* − m × ln (VPD_leaf_), where *g_ref_* is the reference stomatal conductance at 1 kPa VPD (Y-intercept), and the slope m is the sensitivity of *g_s_* [19].

### 4.4. Statistical Analyses

Each parameter from the cavitation curve (*K_max_*, P_50_, P_88_, and S_50_) was subjected to a one-way analysis of variance, with clone as the main factor with four levels (i.e., four clones). Then, we used contrast analysis to test the differences between the northern (SanFer and Romer) and southern (Enlagos) clones. On the other hand, the *g_s_* response to VPD was analyzed separately between provenances and clones using covariance analysis [60]. We tested whether the slope and intercept of the relationship *g_s_* to VPD differed among the provenances and clones. Mean comparisons for mean *g_s_* were carried out with Tukey’s test. All the analyses were performed with the program R (r-project.com) and considered significant at an alpha level of 0.05.

## 5. Conclusions

This study showed that the provenances of *A. chilensis* under study exhibited moderate vulnerability to cavitation and moderately low stomatal sensitivity to VPD. Similarly, genotypes (i.e., clones) from contrasting environmental conditions did not differ in vulnerability to cavitation and stomatal sensitivity to VPD, suggesting that the species might have low genetic variation and resilience to increasing drought and VPD due to climate change. However, further research is needed to study the acclimation of these parameters under drought conditions and in older plants coming from more xeric and wet sites compared with those in our study.

## Figures and Tables

**Figure 1 plants-10-01777-f001:**
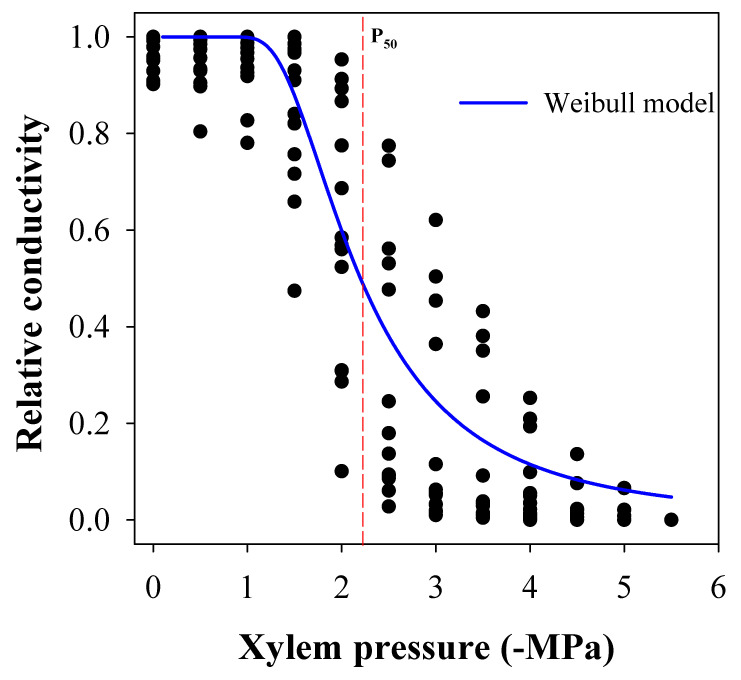
Relative loss of hydraulic conductivity for 2-year-old maqui plants (*Aristotelia chilensis* (Molina) Stuntz).

**Figure 2 plants-10-01777-f002:**
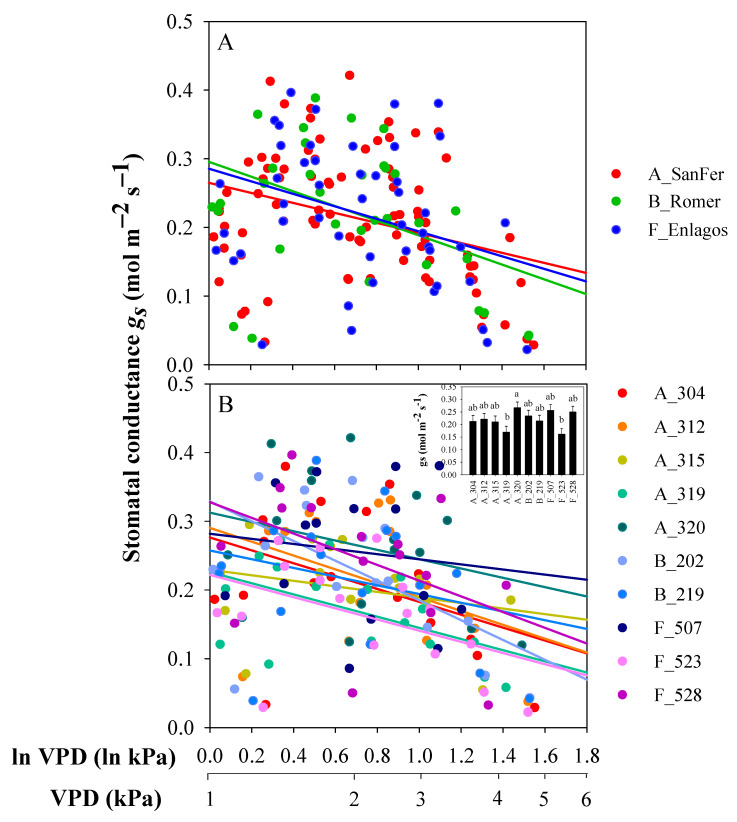
Relationship between stomatal conductance (gs) and the logarithm of VPD in 2-year-old maqui plants (*Aristotelia chilensis* (Molina) Stuntz) by provenance (**A**) and by clone (**B**). The VPD scale is also presented. The inset graph shows the average stomatal conductance by clone, and different letters indicate significant differences (*p*-value < 0.05) between the genotypes based on Tukey’s means comparison test.

**Figure 3 plants-10-01777-f003:**
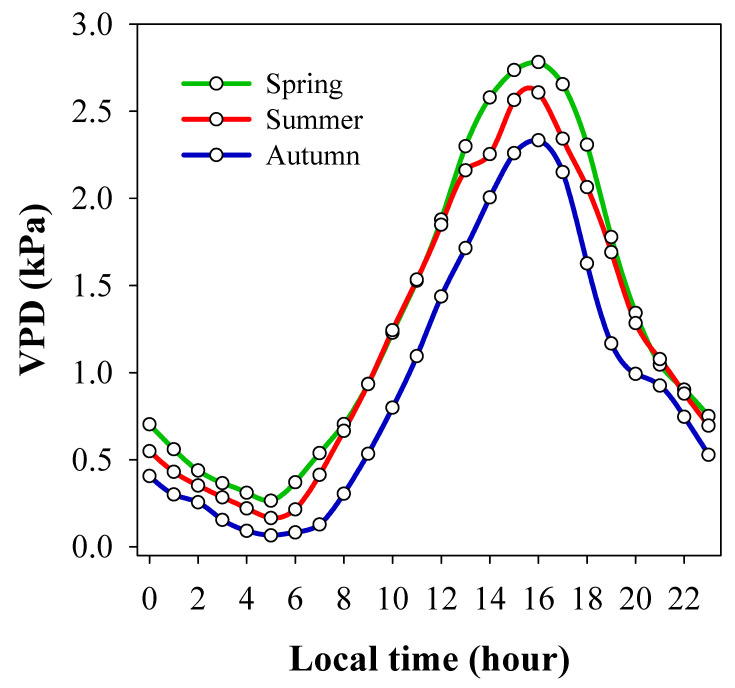
Diurnal variation of vapor pressure deficit (VPD) in the experimental location at the end of spring, midsummer, and the beginning of autumn.

**Table 1 plants-10-01777-t001:** Location, altitude, Köppen climate classification, climate data, and solar radiation for the *Aristotelia chilensis* (Maqui) provenances under study.

Provenance	Latitude (°S)	Longitude (°W)	Altitude (m)	Köppen Classification ^1^	Mean Temperature (°C)	Temperature Min–Max ^2^ (°C)	Precipitation (mm)	Global Radiation (MJ/m^2^)
San Fernando (SanFer)	34°41′	70°50′	530	Csc	14.0	7.0–20.9	552	19.1
Romeral (Romer)	34°57′	70°57′	495	Csb	13.5	6.4–20.6	833	18.5
Entre Lagos (Enlagos)	40°40′	72°33′	165	Cfb	11.4	6.5–16.4	1855	13.7
Nursery location	35°34′	71°22′	275	Csb	13.9	6.7–21.1	869	18.1

^1^ Köepen climate classification, ^2^ Min and Max correspond to Minimum and Maximum temperatures.

## Data Availability

Data is contained within the article.

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
