# Peer review of "Limited Phenotypic Variation in Vulnerability to Cavitation and Stomatal Sensitivity to Vapor Pressure Deficit among Clones of *Aristotelia chilensis* from Different Climatic Origins"

_plants, 2021, doi:10.3390/plants10091777_

Round 1

Reviewer 1 Report

This study tested clones of one species from three locations for variation in stem hydraulic failure and stomatal sensitivity to VPD.  They found little variation between locations of origin.  This is a well conceived and executed study.  The lack of variation was not expected, which decreases the interest in the results, but one can not fault the effort or the results.

There are a few clarifications which would help:

Lines 17 & 83 - should be "contrasting climates"

What is the climate of the nursery location?

Line 159:  "nearly" should be "approximately"

Line 254: "expected" should be "used"

Methods:  In the case where VPD was increased by drying of the air, how much time was allowed for gs to become stable?

The equation relating gs to VPD does not deal with possible temperature differences.  I suspect that temperature might account for some of the gs variation in addition to VPD. 

Reviewer 2 Report

Dear authors,

the aim of this paper is to describe phenotypic variability in P50 and stomatal sensitivity to VPD of Aristotelia chilensis. The study was done on 10 clones. Even though these clones had different origin, the phenotypic plasticity of studied traits was small. I appreciate this type of study, which can show the adaptability potential of particular species compare to interspecies comparison. This type of studies can thus be used in the models predicting species distribution under different climatic scenarios.  Overall, the article brings interesting results, but I think that major revision should be done before it will be accepted.   See my detail comments bellow:

Keywords - Do not use same words as in Abstract.

Line 21 - P50 of "-" 2.2 MPa

Line 21-22 - What type of parameters?

Line 23-24 - Moderately low?- in respect to?

line 26 - Can you describe to possibilities of further research?

line 53 - Not clear connection - Populus tremuloides and poplars?

lie 54-55 - Why do you think it is controversial? These and other paper show that vulnerability to cavitation is species specific.

line 56-58 - Not clear sentence as well as the connection with previous paragraph.

line 58  - ...typically decreases with increase VPD, ....

line 62 - style of reference

line 82-83 - The abbreviation "VPD" was already explained.

line 95-97 - Not clear sentence. Now it is mixture of parameters.

line 101 - VPD - 0.86 to 4.7 MPa or kPa?

line 103 - Important note! It is hard to make some conclusion from your dataset about the relationship between VPD and gs at higher VPD range (more than 3 kPa) as you have done here only few measurements. Can you add more points at higher VPD range? Now, your conclusions are not strongly supported by your data.

line 137-138 - moderate - compare to??

line 147 - Why you use "However" here?

line 153-154 - Important note! The other experiment should not only increase the number of provenances (included northern populations), but it should also increase the dataset. You should measured more individuals per clone and also measurement at higher VPD range). It valid also for vulnerability curves. You should also study drought response of this species.

line 178-179 - Are you sure that Taxodium distichum was considered as a species with low resistance to cavitation?

line 185-200 - Your dataset is influenced by small number of measurement at higher VPD range.

line 204-207 - Iso- and aniso-hydric strategy has it advantage and disadvantage during drought. This is largely depending on drought intensity, duration and frequency.

line 218-219 - Important note! It is not clear how many individuals were used for vulnerability curves and stomatal sensitivity measurement per clone.

line 260 - Important note! It was already published in several papers that in general critical threshold for conifer is P50 and for angiosperm P88. You should add to your results and discussion also P88 values.

line 265 - Leaf stomatal conductance is not shown on Fig. 1. Why stomatal conductance was measured only in the morning?

line 267 - Can you write when you measured in the afternoon?

line 269 - Why stomatal sensitivity to VPD was not measured by gradually increasing VPDleaf (function of the LI 6800)? Then you will have same number of measurement for whole VPD range.

line 279 - parameters cmax, S50 is not explain in Methods.

Fig. 2 - Important note! Can you distinguish clones in your figure? You should show the variability curves per each clones.

Fig. 3 - Important note! Why linear regression is used here? See line 275-276 , where you wrote that you use logarithmic regression.

Reviewer 3 Report

In the present study, Yanez and co-Authors estimated possible changes in xylem vulnerability to embolism of different clones of Aristotelia chinensis. Cuttings were collected from different sites and grown for about one year in a nursery facility.  Results showed no changes in water potential inducing 50% loss of hydraulic conductivity (i.e., P50 values). Moreover, no changes in stomatal sensitivity to VPD were recorded among the different clones. On the basis of these data, Authors attributed low plasticity to the study species.

Honestly, I don’t agree with this conclusion. The experimetal planning was not designed to highlight potential phenotypic plasticity of plant traits, including xylem embolism vulnerability In fact, samples were grown under similar growth condition. And this, in turn, may allow to highilght genotypic traits but not phenotypic variability. In accordance, the Authors cited an interesting study n. 35 in the Manuscript) in which Wortemann et al. showed as seeds collected from different sites showed similar P50 value when grown under the same growth condition. By contrast, differences in xylem vulnerability to embolism were recorded among samples when samples were grown in different sites (i.e., environmental conditions). A similar experimental planning was performed by Corcuera et al ( ref. n.13). Moreover, in the cited studies n. 6, 12 and 14, the measured samples were collected from different growing sites. As widely known, changes in P50 value and, then, in xylem vulnerability to embolism is tightly related to xylem anatomical traits (just as same samples, please see Kaack et al., 2021, New Phytologist, Johnson et al., 2020, Plant Physiology). And these, in turn, are related to growth conditions (i.e., Lemaire et al., 2021, Tree Physiology; Herbette et al., 2020, BioRxiv; Plavcova et al., 2012, Experimental of Botany).

In summary, on the basis of the experimental planning, Authors are able to affirm only that the measured clones of A. chinensis didn’t show differences in xylem vulnerability, neither in stomatal conductance sensitivity to VPD. On this view, sections in the Introduction and the Discussion must be abudandtly improved/changed.

Not at last, measuring of the maximum vessel lenght (MVL) is very relevant to validate the data of vulnerability curve. MVL strongly impacts the vulnerability to xylem embolism (i.e. Ennajeh et al., 2011, Physiologia plantarum; Torres-Ruiz et al., 2017, New Phytologist; Zhao et al., 2020, Physiologia Plantarum). In accordance, the relatively high P50 value may be due to an experimental artefact (i.e., the using samples with open vessels).

Round 2

Reviewer 2 Report

Dear authors,

I have read your revision and I think that the manuscript could be now accepted for publication.

Author Response

Reviewer 2 had no comments to the revised version and accepted it.

Reviewer 3 Report

The manuscript has been improved. Minor revision is requested.

L.85-86: Please specify “of origins clones “ .

L. 98: Attention: the unit of Kmax is wrong and, likely, the value. The unit of hydraulic conductance of the sample is Kg (or mmol) s-1 MPa-1. Please, check/rectify.

L. 258-261: Please, provide the value of the maximum vessel lenght. To avoid artefacts, measured sample must be longer than the maximum vessel lenght.

Author Response

Response to reviewer 3

We appreciate all the comments made by the reviewer. 

Comment 1.- L.85-86: Please specify “of origins clones “ .

Response: Corrected.

Comment 2.- L. 98: Attention: the unit of Kmax is wrong and, likely, the value. The unit of hydraulic conductance of the sample is Kg (or mmol) s-1 MPa-1. Please, check/rectify.

Response: Corrected to the units suggested.

Comment 3.- L. 258-261: Please, provide the value of the maximum vessel lenght. To avoid artefacts, measured sample must be longer than the maximum vessel lenght.

Response: Corrected. The maximum vessel length measured  was 21.3 m, which was expressed in the range form as appear in the revised version.